# Pasteurized Milk Serves as a Passive Surveillance Tool for Highly Pathogenic Avian Influenza Virus in Dairy Cattle

**DOI:** 10.3390/v17101318

**Published:** 2025-09-28

**Authors:** Abhinay Gontu, Manoj K. Sekhwal, Anastacia Diaz Huemme, Lingling Li, Sophia Kutsaya, Michael Ling, Nidhi Kajal Doshi, Maurice Byukusenge, Ruth H. Nissly

**Affiliations:** 1Huck Institutes of the Life Sciences, The Pennsylvania State University, University Park, PA 16802, USA; abhinay@psu.edu (A.G.);; 2Animal Diagnostic Laboratory, Department of Veterinary and Biomedical Sciences, The Pennsylvania State University, University Park, PA 16802, USA; mmk6053@psu.edu (M.K.S.); anastaciadiaz1012@gmail.com (A.D.H.); lul17@psu.edu (L.L.); sophiakutsaya@gmail.com (S.K.); bmaurice@psu.edu (M.B.)

**Keywords:** highly pathogenic avian influenza virus, cattle, genome sequencing, H5N1, milk

## Abstract

The emergence of H5N1 highly pathogenic avian influenza virus (HPAIV) clade 2.3.4.4b in dairy cattle across multiple U.S. states in early 2024 marks a major shift in the virus’s host range and epidemiological profile. Traditionally limited to bird species, the ongoing detection of H5N1 in cattle, a mammalian host not previously considered vulnerable, raises urgent animal and human health concerns about zoonoses and mammalian adaptation. We assessed the feasibility of using commercially available pasteurized milk as a sentinel matrix for the molecular detection and genetic characterization of H5N1 HPAIV. Our aim was to determine whether retail milk could serve as a practical tool for virological monitoring and to evaluate the use of full-length genome segment amplification for extracting genomic sequence information from this highly processed matrix. Our results link HPAIV sequences in store-bought milk to the cattle outbreak and highlight both the potential and the limitations of retail milk as a surveillance window. Together, these findings provide evidence that influenza A virus RNA can be repeatedly detected in retail milk in patterns linked to specific supply chains, with genomic data confirming close relationships with the viruses circulating in cattle.

## 1. Introduction

The historical importance of zoonotic influenza viruses is demonstrated by the rise of several pandemic strains throughout the past century, such as H1N1 in 1918 and 2009, H2N2 in 1957, and H3N2 in 1968 [1,2,3,4]. These events underscore the critical importance of vigilant surveillance for influenza A viruses (IAVs) with zoonotic and pandemic potential [1,5]. One such virus of increasing global concern is the H5N1 highly pathogenic avian influenza virus (HPAIV), particularly clade 2.3.4.4b [6]. Since 2020, viruses of this clade have exhibited sustained geographic spread and a notable capacity for cross-species transmission, including sporadic spillover into human populations and a growing list of susceptible mammalian hosts [6,7].

In early 2024, H5N1 HPAIV clade 2.3.4.4b was detected in dairy cattle across multiple U.S. states, marking a critical shift in the virus’s host range and epidemiology [8,9,10]. Cattle have not historically been regarded as reservoirs for avian influenza viruses, rendering this expansion in host tropism particularly concerning and highlighting the first sustained transmission of H5N1 HPAIV in a mammalian host [11,12]. Multiple spillovers into dairy cattle have now been confirmed [13,14]. The establishment of H5N1 HPAIV in bovine populations raises significant public health concerns, particularly in light of the extensive human–animal interface present in agricultural environments. Although all human cases reported to date have been isolated and without sustained human-to-human transmission, the continued persistence in cattle herds and transmission between animals and farms increases the chance of adaptation in the mammalian host that could enhance risk to humans.

Continual surveillance and genetic characterization of circulating H5N1 HPAIV is essential to identifying mutations that could affect transmissibility, virulence, and pandemic potential. Changes in the surface glycoproteins hemagglutinin (HA) and neuraminidase (NA) can alter receptor binding and release, thereby shaping host range and pathogenicity. Mutations in the non-structural protein 1 (NS1) affect the virus’s ability to evade innate immune defenses, influencing virulence. The polymerase basic protein 1 (PB1) also plays a central role in host adaptation and replication efficiency. Several concerning mutations in HA, NA, NS1 and PB2 were present in the initial samples from the 2024 cattle outbreak [9,15,16], emphasizing the importance of monitoring changes as virus circulates in this host species.

Most studies and surveillance of cattle for H5N1 HPAIV focus on farm-level or bulk tank sampling [8,9,15,17,18,19,20,21]. Less is known about what viral signal persists through processing to the consumer level. The United States Department of Agriculture’s National Milk Testing Strategy (NMTS) is robust at early detection of H5N1 HPAIV in unprocessed milk, helping to identify affected herds [22]. However, NMTS and similar bulk tank and plant silo sampling involves pooled, unpasteurized milk collected before processing and does not represent the end products consumed by humans. H5N1 HPAIV viral RNA has been detected in raw and pasteurized milk [8,10,17,18,23,24,25], but persistence, detection patterns, and genomic data from consumer products are underexplored.

The IAV genome is composed of eight negative-sense RNA segments, each ranging in length from 890 to 2341 nucleotides. Amplification of all eight intact segments can be achieved using a single primer set that anneals to conserved sequences at the 5′ and 3′ termini of each segment [26,27,28]. In recent years, streamlined protocols have been developed for long-read sequencing of such amplified IAV genome segments using relatively low-cost supplies and Oxford Nanopore Technologies devices, facilitating genetic characterization of avian influenza viruses [29,30,31,32,33]. Sequences of intact genome segment molecules from pooled samples can reveal whether complementary mutations co-localize within the same segment or occur separately in different variants of the viral population.

In this study, we explored the potential of commercially available pasteurized milk as a sentinel matrix for the molecular detection and genetic characterization of H5N1 HPAIV. Distinct from traditional epidemiological surveillance, which emphasizes infection prevalence, we longitudinally sampled retail milk from multiple brands and processing plants to characterize influenza A virus detection over time. We also assessed genomic recovery using universal primers and long-read sequencing to evaluate the potential insights that can be drawn via this method.

## 2. Materials and Methods

### 2.1. Sample Procurement

Pasteurized commercial milk samples were obtained in State College, Pennsylvania, over seven months from May to November 2024. The majority were obtained directly from retail store shelves in sealed, tamper-evident containers, and a minority were acquired as aliquots through community crowdsourcing from known donors affiliated with the Pennsylvania State University (Appendix A).

### 2.2. Nucleic Acid Extraction and RNA Purification from Milk Samples

Viral RNA was extracted from milk using procedures based on approaches previously used on dairy cow milk [8,10] and validated for the surveillance of avian influenza virus [34]. A total of 200 µL of undiluted commercial milk was used for nucleic acid extraction. Total RNA was isolated using the MagMAX™ CORE Nucleic Acid Purification Kit (Cat. A32702, Thermo Fisher Scientific, Waltham, MA, USA) according to the manufacturer’s protocol with Xeno RNA, yielding a final elution volume of 90 µL. The extracted nucleic acid was used directly, without dilution, for real-time RT-PCR assays. For sequencing applications, the nucleic acid underwent DNase treatment followed by RNA purification and concentration using the RNA Clean and Concentrator™-25 kit (Cat. R1018, Zymo Research, Irvine, CA, USA). The RNA was concentrated approximately 10-fold and subsequently used for H5 gene sequencing or whole-genome sequencing.

### 2.3. Real-Time RT-PCR Detection of Influenza a Virus

Real-time RT-PCR (rRT-PCR) was performed to detect the Influenza A virus (IAV) matrix (M) gene using a hydrolysis probe-based assay previously validated by USDA for avian influenza virus surveillance [35]. The following previously published primers and probe [35] were used: F25 forward (5′-AGA TGA GTC TTC TAA CCG AGG TCG-3′), R124 reverse primer (5′-TGC AAA GAC ACT TTC CAG TCT CTG-3′) and R124 modified primer (5′-TGC AAA AAC ATC TTC AAG TCT CTG-3′), and F64 probe (5′-/56-FAM/TCA GGC CCC/ZEN/CTC AAA GCC GA/3IABkFQ/-3′), all synthesized by Integrated DNA Technologies (IDT). Each 25 µL reaction contained 4.5 µL of nuclease-free water, 0.25 µL of each primer (20 µM), 0.25 µL of probe (6 µM), 1 µL of VetMAX™ Xeno™ Internal Positive Control—VIC™ Assay (Cat. A29765, Thermo Fisher Scientific), 12.5 µL of 2× RT-PCR buffer, 1 µL of 25× RT-PCR enzyme mix (both from the AgPath-ID™ One-Step RT-PCR Reagents kit; Cat. 4387391, Thermo Fisher Scientific), and 5 µL of RNA template. Reactions were run on the Applied Biosystems (Waltham, MA, USA) 7500 Fast Real-Time PCR System with the following thermal cycling conditions: reverse transcription at 45 °C for 10 min, initial denaturation at 95 °C for 10 min, followed by 45 cycles of denaturation at 94 °C for 1 s and annealing/extension at 60 °C for 30 s. All samples were tested in duplicate. A non-negative cycle threshold (Ct) value for both replicates was considered positive for IAV RNA. All samples demonstrated amplification of Xeno internal positive control.

### 2.4. Real-Time RT-PCR Detection of H5

Detection of the Influenza A virus hemagglutinin (H5) gene was achieved through quantitative real-time PCR (qPCR) using the Avian Influenza Type A (H5) Primers and Probe Set (Cat. No. 10020786, Integrated DNA Technologies, Coralville, IA, USA), in combination with the Avian Influenza Positive Sample Control (Cat. No. 10020808, Integrated DNA Technologies) and the PrimeTime™ One-Step 4× Broad-Range Master Mix (Cat. No. 10011744, Integrated DNA Technologies). Each 20 µL reaction consisted of 1 µL primer/probe mix, 1 µL of positive control or sample RNA, 5 µL of master mix, and 8 µL of nuclease-free water. Amplification was carried out on an Applied Biosystems 7500 Fast Real-Time PCR System with the following cycling parameters: reverse transcription at 50 °C for 15 min, initial denaturation at 95 °C for 3 min, followed by 40 cycles of 95 °C for 15 s and 60 performed on an Applied Biosystems 7500 Fast Real-Time PCR System using the following cycling parameters: reverse transcription at 50 °C for 15 min, initial denaturation at 95 °C for 3 min, followed by 40 cycles of 95 °C for 15 s and 60 °C for 1 min. Fluorescence signals were collected using FAM for the H5 target and VIC for the internal control. All samples were analyzed in duplicate, and those with a non-negative cycle threshold (Ct) value of at least one replicate were classified as positive for H5 RNA.

### 2.5. Whole-Genome Amplification and Sequencing

Whole-genome sequencing was conducted following the protocol published by Thielen [36] based on universal IAV primers [27]. Modifications were made only to the RNA extraction and concentration procedures, as described in the methods section above. Sequencing was performed on a MinION Mk1c instrument (Oxford Nanopore Technologies, Oxford, UK) with R10.4.1 flow cells, following the manufacturer’s protocol. Basecalling was conducted using Dorado v1.1.1 (https://github.com/nanoporetech/dorado, accessed on 8 August 2025) with the super-accurate (sup) model. Sequencing data were deposited to NCBI under BioProject ID PRJNA1312032.

### 2.6. Genomic Assembly and Phylogenetic Analysis of Whole-Genome Sequences

Whole-genome sequencing data were processed to generate consensus sequences using reference A/Bovine/Texas/24-029328-01/2024. Raw FASTQ files were aligned to a reference genome using the BWA-MEM v0.7.19 [37]. The resulting SAM alignments were converted to BAM format, sorted, and indexed with Samtools v1.22 [38]. For each sample, coverage statistics were calculated to evaluate genome-wide read distribution and completeness. Variant calling was performed using BCFtools v1.22 [39], which generated pileup files and subsequently called variants to produce VCF files containing single-nucleotide polymorphisms (SNPs) and indels relative to the reference genome. To reconstruct sample-specific consensus sequences, iVar [40] was used to generate consensus FASTA sequences for each genome segment.

Phylogenetic analysis was performed on the neuraminidase (NA) gene from three samples, S154, S155, and S156, and hemagglutinin (HA) from S155 and S156 of H5N1 avian influenza viruses. A total of 42 NA sequences and 52 HA sequences obtained from the GISAID EpiFlu database (https://gisaid.org, accessed on 21 August 2025) were included. The multiple sequence alignment of the sequences was performed using MAFFT v7.52 [41]. A maximum-likelihood phylogenetic tree was constructed with FastTree v2.1.11 [42]. The resulting tree was visualized on the iTOL platform v7 (https://itol.embl.de, accessed on 22 August 2025) [43].

### 2.7. Variant Normalization and Annotation

Raw VCF files were normalized using bcftools norm with the reference genome to split multiallelic sites, followed by indexing using bcftools index. Functional annotation of variants was performed using SnpEff v5.2 [44]. Post-processing included custom scripts for variant filtering and amino acid name extraction.

### 2.8. Amplification and Sanger Sequencing of H5 Gene

Reverse transcription and amplification of the H5 gene were performed using the SuperScript™ IV One-Step RT-PCR System with ezDNase™ Enzyme (Cat. No. 12595025, Thermo Fisher Scientific) following the approach described by Suarez et al. [45]. The reaction mixture (25 µL total volume) contained 2.5 µL of concentrated RNA, 12.5 µL of the 2× reaction master mix, 2.5 µL each of forward primer H5+146EA (5′-GTT ACT GTT ACA CAT GCC CA-3′, 5 µM) and reverse primer H5−1347EA (5′-AGT TCA GCA TTA TAA GTC CA-3′, 5 µM), 0.25 µL of the RT mix enzyme, and 4.75 µL of nuclease-free water. Thermal cycling was carried out under the following conditions: reverse transcription at 55 °C for 10 min, initial denaturation at 98 °C for 2 min, followed by 40 cycles of denaturation at 98 °C for 10 s, annealing at 57.2 °C for 10 s, and extension at 72 performed under the following conditions: reverse transcription at 55 °C for 10 min, initial denaturation at 98 °C for 2 min, followed by 40 cycles of denaturation at 98 °C for 10 s, annealing at 57.2 °C for 10 s, and extension at 72 °C for 1 min. A final extension was performed at 72 °C for 5 min, followed by a hold at 4 °C. A ~1200 bp amplicon was obtained. The H5 amplicons were treated with ezDNase and submitted to the Genomics Core Facility at the Huck Institutes of the Life Sciences, The Pennsylvania State University, for Sanger sequencing.

## 3. Results

### 3.1. Influenza a Viral RNA Detection in Pasteurized Dairy Milk

A total of 214 pasteurized milk samples were obtained from May to November 2024. These samples were packaged at dairy processing plants from 12 states and were sold in retail stores located in State College, Pennsylvania. The largest proportion of samples were packaged in Pennsylvania (41.1%), followed by New York (19.2%), Colorado (13.1%), and Michigan (9.8%). Each remaining state encompassed less than 5% of the total processed samples (Table 1; Appendix A). The samples represented 38 different brand names and were packaged across 40 different processing plants.

Following the initial screening of milk samples for the IAV matrix gene by real-time RT-PCR, subsequent collections focused on brands and processing plants that had previously tested positive. Samples from five brands (G, K, AO, A, AT), processed in five plants (2, 3, 5, 8, and 30) across Michigan, Colorado, New York, Virginia, and Missouri, were found to be positive. In total, 47 samples tested positive for the IAV matrix gene. Most (85%) matrix-gene positive samples were also positive for IAV H5 gene via real-time PCR (Appendix A). IAV was detected in all milk fat compositions examined, including whole, 2%, 1%, and skim milk.

Multiple milk products sold under the same brand name tested positive, such as products with different milkfat contents or packaged as either fresh or shelf-stable (Table 2). Certain products of brands A and AO were packaged at the same plant, and these products from both brands consistently tested positive; brand AT was also packaged at one of these plants and tested positive. Additionally, for two brands, we detected IAV in milk products sold under the same brand but packaged at different plants.

By following initially positive samples and selectively collecting additional samples from specific brands and processing plants, we monitored IAV detection over time based on sell-by dates. Four brands, processed in four distinct plants (de-identified to preserve anonymity), were tracked for temporal patterns (Figure 1):Brand G (Plant 2): Positive samples persisted up to 18 weeks before testing negative.Brand K (Plants 3 and 5): Initial samples were positive through week 3, then consistently negative.Brand AO (Plant 8): Samples remained positive until week 8; no further samples were available.Brand A (Plant 8): Positive through week 10 before transitioning to negative, enabling follow-up from Plant 8.

This approach allowed a proxy longitudinal analysis of IAV persistence in milk from specific brands and plants, providing temporal insights despite limitations in sample availability.

### 3.2. Influenza a Virus Whole-Genome Sequencing from Retail Milk

#### 3.2.1. Sequencing Depth and Coverage

Whole-genome amplification was attempted on viral RNA from 10 samples. RT-PCR products using influenza A virus universal primers were obtained and sequenced for 3 of these samples, all from brand AO milk processed at a plant in Colorado. This approach resulted in 60 to 87 percent genome coverage, with 5 or more segments of each specimen fully sequenced at >75× mean depth (Table 3, Appendix A).

#### 3.2.2. Phylogenetic and Lineage Analysis

Phylogenetic trees were constructed for the neuraminidase (NA) and hemagglutinin (HA) genes using whole-genome sequences generated from S154, S155 and S156, together with publicly available U.S. sequences collected between 1 March and 21 August 2024 (Figure 2). The three NA sequences clustered within H5N1 HPAIV clade 2.3.4.4b, forming a strongly supported monophyletic group/common ancestor, indicating high genetic similarity and possible epidemiological linkage. Full-length HA gene sequences available from our samples also clustered with genotype B3.13 clade 2.3.4.4b H5N1 sequences, with close relationships to sequences from Colorado cattle (Figure 2). Although sample S154 lacked next-generation sequence of the HA gene, Sanger sequencing of a H5-specific amplicon (reference) confirmed this sample also contained B3.13 clade 2.3.4.4b H5N1 genetic material (Appendix A).

#### 3.2.3. Variant Analysis

Variant analysis revealed one or more amino acid changes in the retail milk samples compared with the reference strain sequence (Table 4, Appendix A), several of which have been previously described. The consensus sequences from S154 and S156 had the A255V substitution in PB2. Over 95% of reads from S155 and S156 and more than 48% of reads from S154 contained a R21Q mutation in the RNA-binding region of NS1. Sample S156 additionally contained E249G and G673D substitutions in PB2. Variant analysis also indicated a premature stop codon at HA position 504 in S156; however, the depth of coverage at this position (71×) was relatively low compared with other substitutions detected (range 107× to 893×). Sample S154 contained V655A in the consensus sequence of PB2, and a minority of reads (24.5%) encoded an E71G substitution in NS1. The consensus sequence of S154 (>93% of reads) also encoded previously uncharacterized amino acid substitutions S261G and M321T in PB1 and I443T in NA.

## 4. Discussion

This study evaluated the utility of commercially available pasteurized milk as a sentinel matrix for the molecular surveillance and genetic characterization of H5N1 HPAIV, particularly in the context of its recent emergence in U.S. dairy cattle. By tracking IAV in retail milk over time, we identified brand- and plant-associated patterns of detection. We recovered outbreak-related sequences, including mutations with potential relevance to mammalian adaptation. Our findings demonstrate that viral genetic material can persist through commercial dairy processing and remains detectable using real-time RT-PCR. The recovery of partial to near-complete HPAIV genomes from retail milk indicates that processed fluid milk can retain analyzable levels of viral RNA closely related to viruses from clinically affected cattle. Collectively, these findings support the potential use of pasteurized milk as a practical, non-invasive sampling matrix to complement other H5N1 HPAIV surveillance strategies.

The association of specific milk brands and processing plants with viral RNA detection in our dataset suggests persistent sources of infection within dairy supply chains. Dairy industry networks often result in raw milk being processed and packaged in a different state than the cow from which it was produced. Previous studies likewise reported viral RNA in dairy products from plants located in states without reported cattle infections [17,25,45,46]. Consistent with this, we detected H5N1 HPAIV RNA in three states with no reported detection in cattle, New York, Missouri, and Virginia. These observations underscore the importance of judicious interpretation of retail dairy testing results and the need for timely reporting to regional (state) and federal animal health and food safety authorities to facilitate investigation of contamination sources.

The ability to assign these sequences to known HPAIV clades demonstrates how this approach may complement field-based surveillance, particularly in contexts where direct sampling from animals is impractical or limited by logistical constraints. Phylogenetically, our sequences were very closely related to other H5N1 HPAIV strains from cattle. The uniformly short branch lengths and minimal pairwise distances imply that these viruses share a very recent common ancestor, consistent with a spillover event into cattle rather than long-term adaptation within herds. Despite these close relationships, several major consensus-level and minor variants were detected, demonstrating that the universal primer segment amplification approach for recovering genomic material allows monitoring of mutations relevant to increased pandemic potential.

Both NS1 and PB2 play critical roles in IAV host restriction and adaptation, so it is vital to monitor for variations in these proteins. H5N1 HPAIV genetic material in our milk samples contained several substitutions previously associated with mammalian adaptation or host-specific RNA or protein interactions [46,47,48,49,50,51,52]. The R21Q substitution in NS1 has been identified in other specimens associated with H5N1 in dairy cattle and has been noted as a significant feature of mammalian-origin H5N1 sequences [47,53]. The E71G substitution in NS1 is associated with the conformational states of the protein with potential impacts on interactions with host RNA and proteins [48]. A255V [46] and E249G [53] substitutions in PB2 have been observed in other specimens associated with the H5N1 HPAIV outbreak in U.S. cattle. The latter is associated with enhanced binding and virulence in human respiratory cells [49]. G673D in PB2 has also been described in a H1N1 IAV present in a patient co-infected with H7N9 avian influenza virus [50]. In certain non-H5 avian influenza viruses, V655A in PB2 has been associated with higher replication [51] and compensates for reduced polymerase activity in mammalian cells [52].

However, it is imperative to consider sequence variants from retail milk carefully. Bulk tank milk, the starting material for retail milk, represents pooled specimens from multiple animals, which introduces inherent limitations for genomic analysis. In particular, the presence of milk from different individuals increases the likelihood of detecting mixed viral populations, including multiple strains, which may complicate variant calling and downstream interpretation. Additionally, while Oxford Nanopore Technologies sequencing has historically been associated with lower accuracy compared to short-read platforms, recent improvements in flow cell chemistry and basecalling algorithms (e.g., R10.4.1 and Dorado sup) have significantly enhanced data quality. Nonetheless, caution is warranted when interpreting consensus-level variants, and validation through complementary methods remains important.

The utility of retail milk testing for H5N1 HPAIV surveillance has several limitations. Retail milk is temporally lagged from the time of milk production in the cow. While some pasteurized fluid milk reaches consumer shelves within 48 h of milking, advances in pasteurization and packaging technology also allow pasteurized milk to remain fresh for several months. After packaging, some plants store milk containers for weeks before shipping. Thus, retail milk surveillance is not able to detect affected supply nodes in real time. Additionally, milk from multiple herds is commonly combined before being processed and packaged. The pooled nature of retail milk obscures individual farm-level signals. Together, in many instances, retail milk surveillance is an insufficient approach to rapid or comprehensive detection of affected herds.

While adequate for phylogenetic analysis, the success rate of our sequencing approach was limited (3 incomplete whole-genomes from 10 attempted samples). Complete coverage of the shortest three genome segments (NA, NP, and NS) for all three samples suggests that incomplete sequencing or failed amplification of longer segments was compromised due to nucleic acid degradation in the original specimen. However, full-length reads of even the longest segments were obtained for certain samples (Appendix A), indicating that intact viral genomic material may be obtainable from pasteurized, homogenized milk. The samples from which we could obtain sequence were purchased from the same store on a single day, which may suggest that handling of the dairy products during commercial transportation can influence viral RNA stability.

The presence of H5N1 RNA in pasteurized milk does not imply the presence of infectious virus, but it does point to extensive viral circulation at the production source. This observation aligns with current knowledge regarding H5N1 HPAIV tropism for bovine mammary tissue [19,54]. While pasteurization is designed to inactivate pathogens [23,55,56,57], our findings indicate that it does not eliminate detectable viral RNA, nor does it necessarily fragment IAV genome segments. Thus, pasteurized milk may serve as a valuable matrix for monitoring viral genetic material, allowing for the detection of specific clades or lineages over time or across different regions.

## 5. Conclusions

Milk is a complex biological matrix, and post-processing steps such as pasteurization are known to eliminate infectious viral particles and degrade viral RNA [55,56,57,58]. While this degradation is beneficial for food safety, it poses a challenge for intact-segment analysis. Viral genome amplification using primer sets producing tiled amplicons of retail milk-derived RNA have yielded greater coverage and higher success rates than our intact-segment amplification approach [17,46]. However, the short-reads of the tiled amplicon strategy create challenges for virion-level deconvolution and, although unlikely, recombination. Ideally, a combined approach using both tiled and intact-segment amplicons would recover genetic information held both in degraded viral RNA and whole segments present in retail milk. The present study confirms that intact-segment amplification is indeed possible from such samples.

Shortly after our study sampling period concluded, the U.S. initiated routine screening of unprocessed, pre-market milk in December 2024 under its National Milk Testing Strategy (NMTS), utilizing RNA extraction and real-time RT-PCR methods closely aligned with those used in our study. Although retail milk testing may not address concerns of broad national scale, it could be an important approach regionally, such as in counties, state/provinces, or countries where retail milk represents the local dairy population. Future efforts should focus on developing a robust surveillance pipeline to assess the feasibility of using retail dairy products for monitoring H5N1 HPAIV. This study and important related approaches from groups in Minnesota, Massachusetts, Michigan, and Canada guide a framework for performing analysis of milk for H5N1 [17,24,46,59].

This study demonstrates that retail pasteurized milk can serve as a practical matrix for the passive molecular surveillance of H5N1 HPAIV. Spillovers of other influenza A viruses into cattle have also been documented, although it is unknown if such infections result in viral shedding in milk [60,61,62,63,64,65,66,67,68,69,70]. If so, our sequencing approach could potentially be applied to genetically characterize other IAVs in pasteurized milk since the approach uses IAV universal primers for segment amplification. Over time, pasteurized milk testing studies can help evaluate effectiveness of efforts to interrupt viral spread between dairy herds. It can be used to identify viral signatures that escape detection by other surveillance approaches. The detection of viral RNA from a widely distributed consumer product offers a complementary approach to bulk tank and plant silo milk surveillance.

## Figures and Tables

**Figure 1 viruses-17-01318-f001:**
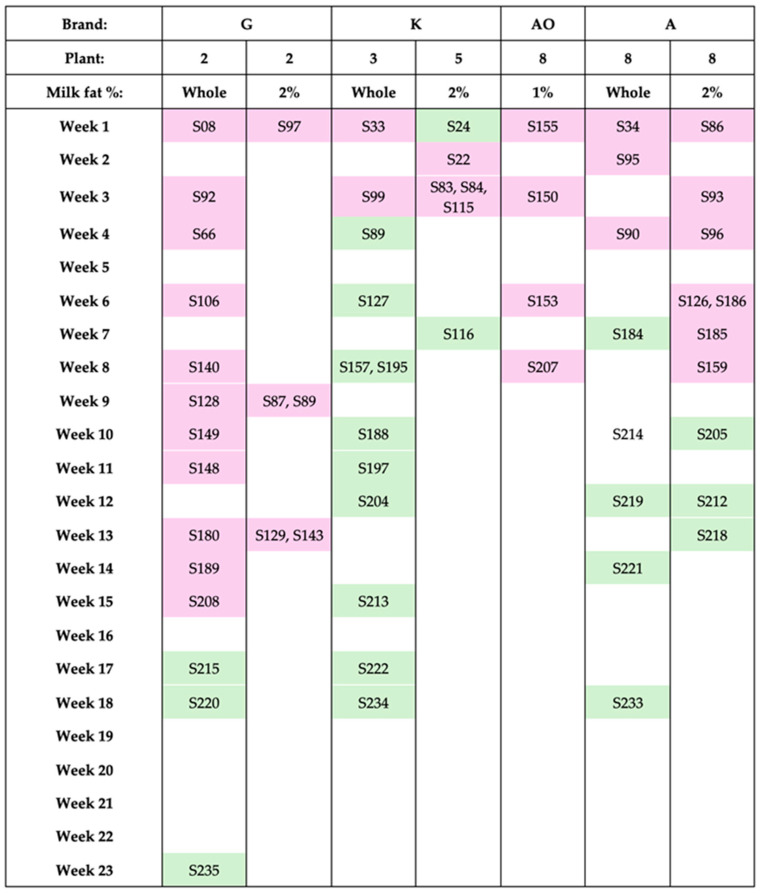
The longitudinal detection of IAV viral RNA in milk products. Sample(s) tested during each period are indicated by numbers. Pink-shaded samples numbers indicate IAV detectable; green-shaded sample numbers indicate IAV undetectable. Unshaded entries indicate no sample was available to test in the period.

**Figure 2 viruses-17-01318-f002:**
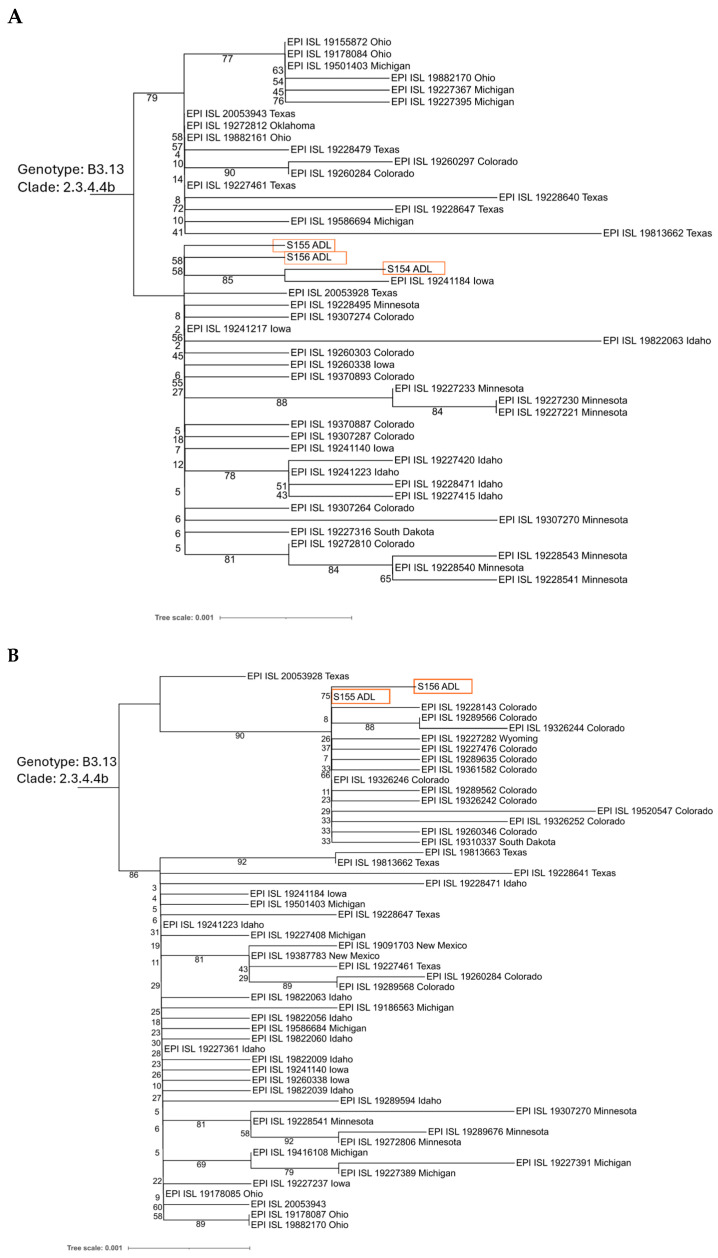
A phylogenetic tree of the H5N1 NA and HA genes generated in this study. The phylogenetic tree was constructed using neuraminidase (**A**) or hemagglutinin (**B**) gene segments extracted from the H5N1 whole-genome sequences generated in this study (highlighted in red), and publicly available sequences from isolates collected between 1 March 2024, and 21 August 2024 (sourced from GISAID). Multiple sequence alignment was performed using MAFFT v7.52, and the trees were visualized using iTOL v7. Bootstrap support values are indicated at key nodes.

**Table 1 viruses-17-01318-t001:** Milk testing results.

Plant State	Plants Tested	Brands Tested	Samples Tested (% of Total)	IAV Positive Samples
CA	1	1	1 (0.47%)	0
CO	1	2	28 (13.08%)	20
IN	1	1	4 (1.87%)	0
MD	1	1	1 (0.47%)	0
MI	2	2	21 (9.81%)	17
MN	2	1	6 (2.80%)	0
MO	1	2	4 (1.87%)	3
NJ	1	1	8 (3.74%)	0
NY	6	7	41 (19.16%)	3
OH	2	2	6 (2.80%)	0
PA	21	16	88 (41.12%)	0
VA	1	1	6 (2.80%)	4
Total	40	38	214 (100.00%)	47

**Table 2 viruses-17-01318-t002:** Brand/plant relationships and products.

Brand	Plant	Product Characteristics				
		Refrigerated				Shelf-Stable
		Skim	1%	2%	Whole	Whole
G	2	x		x	x	
K	3				x	x
K	5			x		
AO	8	x	x	x	x	
A	8		x	x	x	
A	30			x	x	
AT	30		x			

x sample of this type tested positive for presence of influenza A virus by real-time RT-PCR.

**Table 3 viruses-17-01318-t003:** Whole-genome sequencing results.

Specimen	Genome Coverage	Full-Length Segments	Segment Sequence Coverage (Percentage)							
			1 (PB2)	2 (PB1)	3 (PA)	4 (HA)	5 (NP)	6 (NA)	7 (MP)	8 (NS)
S154	87%	7	100	100	100	0	100	100	100	100
S155	60%	5	20	41	0	100	100	100	100	100
S156	65%	5	100	40	19	100	0	100	100	100

**Table 4 viruses-17-01318-t004:** Mis-sense single nucleotide variants compared with reference strain A/Bovine/Texas/24-029328-01/2024.

		Read Depth at Site (Variant Frequency)		
Encoded Protein	Substitution	S154	S155	S156
NS1	R21Q	326 (48.8%)	321 (96.3%)	324 (95.4%)
NS1	E71G	429 (24.5%)	n/a	n/a
NA	I443T	830 (95.2%)	n/a	n/a
PB1	S261G	631 (99.5%)	*	*
PB1	M321T	676 (93.6%)	*	*
PB2	E249G	n/a	*	107 (97.2%)
PB2	A255V	619 (96.1%)	*	107 (100.0%)
PB2	V655A	893 (89.5%)	*	n/a
PB2	G673D	n/a	*	112 (97.3%)

* the segment from this specimen has <100% coverage and was excluded from variant analysis. n/a not applicable; the indicated substitution was not detected in this specimen.

## Data Availability

The original sequencing data presented in the study are openly available in NCBI under BioProject ID PRJNA1312032. Other original contributions presented in this study are included in the article/Appendix A. Further inquiries can be directed to the corresponding author.

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
