# Peer review of "Pasteurized Milk Serves as a Passive Surveillance Tool for Highly Pathogenic Avian Influenza Virus in Dairy Cattle"

_viruses, 2025, doi:10.3390/v17101318_

Round 1
Reviewer 1 Report
Comments and Suggestions for Authors
The manuscript is a timely addition to the knowledge base. The data presented well and not over-interpreted. I do not have specific questions about the work but are providing some overarching comments for consideration.
- This research is not novel as the National milk testing program has been active for months, something mentioned by the authors. It is unclear why the same testing protocols were not utilized by the research team, something that should be addressed in the discussion.
- Yes, MinIon does technically generate long read sequence.....longer than Illumina but I and others would likely not classify it as robust, high fidelity data. Further complicating this, the statement found on page two of the manuscript, "These advances enable individuals with minimal molecular biology training to perform whole-genome sequencing". This is not really a great point as although we could, we would not want untrained technicians analyzing this data.
- RNA quality and pasteurization. Stated in the manuscript, pasteurization may degrade RNA. The methods do not indicate if a complete copy of viral RNA was required for sequencing. It may have been buried in a reference but it should be clear to the reader if this is the case. Average read length should be presented when stating that the processing of dairy does not degrade or impact AIV RNA. The RNA segments of AIV are not particularly long so perhaps the authors dive into this a bit more. I assume that single strand, non-segmented genomes are more easily degraded by pasteurization than shorter, segmented RNA such as is found in AIVs.
- Virus viability. Some attempt should have been made to recover live virus. We already knew/know that AIV is in the milk supply at the farm and silo level.....so why not at the store. The authors mention that they believe that the RNA is coming from virus that is not viable but the data are not presented or available.
- Know your sample source. Please recognize your loss of control over samples submitted by crowdsourcing aliquots of milk across the US. The sources of your non-PA samples are questionable. This should be addressed in the discussion. The intentions of your cooperators are unknown.
- Historical significance. Cows do get infected with influenza A viruses that are not of the H5N1 subtype. We do not know if these infections have resulted in viral shedding in milk. We also do not know if previously reported unresolvable mastitis cases were due to H5N1 infections simply because these animals were not tested for the presence of AIV. There is a good deal of uncertainty pertaining to the arrival of AIV in bovine species. This should be mentioned.
Reviewer 2 Report
Comments and Suggestions for Authors
The authors present a study where they analyzed retail milk for the presence of influenza virus, in particular the H5N1 subtype, and assessed whether retail milk could be used as a passive surveillance tool and virological monitoring. The authors were able to relate the presence of influenza RNA to company brands, the type of milk produced, the states the farms were located, and the states where the milk was packaged. The authors also assessed WGS methods for sequence analysis of the viral genes and the limitations of these analyses. Overall, the authors determined that retail milk would be a suitable tool for passive surveillance.
Recommendations:
Table 4: To better assess the results being presented in section 3.2.3, authors should incorporate the depth of coverage and frequency values into Table 4.
The authors did not discuss the mutations they discovered in any depth during the discussion. I would suggest expanding the discussion to cover the mutations presented or alternatively, include a column in Table 4 providing a brief description of the known functions.
